# communications
# engineering

# Coronavirus-like all-angle all-polarization broadband scatterer

Anna Mikhailovskaya [1,4 ✉], Dmytro Vovchuk[1,4], Konstantin Grotov[1,4], Denis S. Kolchanov [1], Dmitry Dobrykh[1], Konstantin Ladutenko[1], Vjačeslavs Bobrovs [2], Alexander Powell[3], Pavel Belov[1] & Pavel Ginzburg[1]

Creeping waves traveling around a volumetric electromagnetic scatterer provide a significant contribution to its radar cross-section. While quite a few efforts were devoted to suppressing creeping waves as a part of radar countermeasures, here we utilize specially engineered creeping waves to our advantage to create broadband, all-angle, and polarization scatterers. Metalized spherical surfaces, patterned with corona virus-like spikes are designed to provide a broadband constructive interference between the specular reflection and creeping waves, elevating the scattering cross-section. The demonstrated miniature corona scatterers utilize both resonant cascading phenomena and traveling wave interference to tailor electromagnetic interactions, outperforming a resonant dipole in terms of amplitude and bandwidth quite significantly. Our experimental samples are fabricated with an additive manufacturing technique, where a 3D-printed plastic skeleton is subsequently metalized. Micron-thick layers allow governing electromagnetic interactions as if the entire object was made of solid metal. Lightweight, all-angle, all-polarization, and broadband compact scatterers such as these, reported here, have numerous applications, including radar deception, electromagnetic beckoning, and many others.

[1] School of Electrical Engineering, Tel Aviv University, 69978 Tel Aviv, Israel. [2] Institute of Telecommunications, Riga Technical University, Riga, Latvia. [3] Department of Physics and Astronomy, University of Exeter, Exeter EX4 4QL, UK. [4] These authors contributed equally: Anna Mikhailovskaya, Dmytro Vovchuk, Konstantin Grotov. ✉email: anna2@mail.tau.ac.il

Electromagnetic scattering on wavelength-comparable objects is governed by interference phenomena. While only a few shapes (e.g., spheres, ellipsoids, and infinite cylinders) have closed-form analytical solutions[1,2], analysis of more complex structures requires applying numerical tools. However, the design and optimization of large-scale structures, encompassing wavelength-comparable features, demand significant computational efforts, which motivate the development of approximations and intuitive approaches. Creeping waves are a representative example, which comes to describe a constructive interference phenomenon, resulting in an increase of a scattering cross-section (SCS). The Mie resonances of a sphere can be approximated by considering two channels—(1) specular reflection and (2) a wave creeping around the object. If the sphere's circumference is roughly equivalent to the integer number times the free-space wavelength, interference between these two channels can significantly enhance the scattering cross-section.

While describing Mie resonances with creeping waves has primarily theoretical significance, addressing complex shapes with this approach can be quite beneficial. A representative example here is a countermeasure against L-band (low GHz) radars, used for long-range air surveillance[3]. Aircraft fuselage and engine turbines, typically having rounded form factors, provide significant contributions to radar returns at the relevant frequency range. As a result, those shapes are to be avoided in stealth realizations along with other designs, aiming at creeping wave suppression.

However, there are many other scenarios, where enhancing radar cross-section has value. Electromagnetic chaff for radar deception and radiofrequency beacons to increase radar visibility are examples where high radar scattering is desirable. Designing structures for those purposes and other relevant applications typically falls into two extremes—(1) subwavelength resonators and (2) large-scale objects, which responses are governed by physical optics. In the first scenario, resonant conditions allow for achieving significant radar cross-sections. Furthermore, cascading multiple resonances enable reaching super-scattering regimes, where electrically small geometries exhibit superior scattering performances, prevailing those of a single dipolar resonance both by means of maximal cross-section and the nonvanishing bandwidth[4–17]. However, in a vast majority of cases, those structures are significantly biased to a certain polarization and suffer from a reduced bandwidth. While there are quite a few designs of large-scale structures, obeying different engineering constraints, corner reflectors are among the most efficient implementations, demonstrating high radar cross-sections. This fact makes them a preferable choice for radar calibration routines, marine reflectors, and many others. Furthermore, those structures are extremely broadband and work with any incident polarization. However, an efficient operation of a corner reflector requires the structure to encompass ~10 wavelengths in each direction, making devices bulky in the case of low-GHz applications. For example, corner reflectors are not a choice for disposable radar chaff for those frequencies.

Here we propose new electromagnetic scatterers, taking advantage of both resonant and physical optics phenomena. Interplaying between these design approaches allows for achieving both strong scattering from an electrically small (wavelength-comparable at the higher frequencies) structure and significant polarization-independent bandwidth. Our optimization starts from a smooth metallic sphere, which obviously has a broadband all-angle-all-polarization response. Next, this geometry evolved to more complicated shapes, where an array of spikes is added to the spherical surface. This pattern is designed to control creeping waves, traveling along the circumference. Variable dimensions of spikes provide multiple resonant conditions along the entire bandwidth, which allows elevating the scattering cross-section while keeping the operation broadband. Our full-wave numerical analysis and detailed experimental investigations, including near-field scans, support those claims. Experimental samples are fabricated using several methods, where a thin layer of metal is deposited on 3D-printed plastic skeletons. As a result, the obtained structures are extremely low-cost and lightweight, which makes them attractive in numerous practical applications, including radar chaff.

## Result and discussion

**The electromagnetic design and structure's optimization.** Numerical optimization tools keep advancing with ever-growing computational abilities. Topology optimization, where an initial structure is deformed toward maximizing a pre-defined cost function, was found beneficial on many occasions, e.g.,[18–20]. An initial guess of the design is quite important, as it allows convergence to an optimal solution faster, though it is not rigorously guaranteed. Here we start with a sphere, which is a wavelength-comparable efficient scatterer, insensitive to an angle of incident and polarization. We then functionalize the structure's surface with an array of spikes (details in the chapter Methods). Their pyramidal geometry virtually creates a series of gratings with a variable period, depending on an effective elevation above the surface of the sphere (Fig. 1). This pattern can be shown to support resonant creeping waves, traveling along the circumference. Utilizing subwavelength texturing to generate surface waves is an established technique in microwave research, e.g.,[21–24]. Pyramidal structures, made of absorptive materials, are used to cover anechoic chamber walls and ensure all-angle all-polarization scattering suppression. While these descriptions follow an established intuitive logic, they solely suggest probable configurations for the optimization, and further analysis be performed with a full-wave numerical simulation.

Our cost function, subject to maximization, is the total scattering cross-section in the 6–16 GHz frequency range. An additional constraint is keeping the spectral response flat (3 dB variation over the bandwidth). The constraint on the geometry is the radius of a virtual sphere, enclosing the structure. Here we choose radius $R = 20$ mm, which is rather arbitrary.

The best two results from the test set were selected for a detailed comparative analysis. Both configurations have $N = 98$ spikes with (width, height) equal to (50,50) and (80,50) which correspond to (1.5,11) and (2,11) mm, respectively. Scattering spectra are assessed versus the performances of a smooth metal sphere (20 mm radius) and a resonant half-wavelength dipole (16 mm length with the resonant frequency 8.5 GHz) (Fig. 1b). Corona scatterers (98,50,50) and (98,80,50) demonstrate a doubling (more accurately, 1.98 and 2.03 times) of the scattering cross-section, compared to that of a sphere, and surpassing that of the resonant dipole by factors 8 and 8.2, respectively. Furthermore, the scattering bandwidth of corona scatterers is 3 times broader than that of the dipole (a half-width maximum over a smoothed spectrum was used).

To reveal the all-angle all-polarization performance of the scatterer, a set of numerical simulations has been done. The results are summarized in Supplementary Note S1. Minor variations, not exceeding a few % along the entire spectrum, were observed. The experimental investigations also did not reveal any angle and polarization dependence of scattering, which is rather expected in cases when multiple spikes are equally distributed on the surface.

The height of the spikes has a significant impact on scattering characteristics. Detailed analysis is shown in Supplementary Note

S5. While elongation of spikes increases scattering at lower frequencies, it degrades the performance at shorter wavelengths and the overall bandwidth (compared to a smooth sphere) drops. The configuration, considered hereinafter, compromises between the scattering cross-section and the bandwidth.

**Creeping waves**. While numerical optimization provides a layout of structures with optimized performances, it does not reveal the physical mechanisms, underlying the performances. To support the claim on creeping waves' contribution, a time-domain analysis must be performed. When illuminated with a short pulse, the structure should exhibit a split response—the first return comes from a specular reflection, while the second one is delayed, as it results from a longer travel around the circumference (Fig. 2a, insets on top). Creeping waves on several common structures were analyzed for calibrating the setup and the results are summarized in Supplementary Note S4.

To demonstrate the contribution of creeping waves on the corona scatterer, we performed a numerical analysis. A 0.25 nsec Gaussian pulse was launched and the response during 20 nsec

time frame was observed. The carrier frequency is 6 GHz. Figure 2a (upper inset) demonstrates the scenario—the incident pulse crosses a monitor and then interacts with the target. The absolute value of the field amplitude is presented (the pulse encompasses 2 oscillations). The first nanosecond of the interaction was found to contain the main features. Time-dependent backscattering on a smooth sphere and corona scatterer (98,80,50) are compared in Fig. 2a. Two main features are visible—a specular return and a creeping wave, corresponding to a single path along the circumference. The specular return of the corona scatterer is delayed by 0.05 nsec with respect to the wave, traveling on a smooth sphere. This group velocity delay comes from a patterned surface and is rather expected. Additional significant delays can be achieved in the future with a more complex surface morphology (e.g., with tailoring spoof plasmons[24,25]). The creeping wave amplitude on the corona scatterer is significantly (more than twice) stronger than on the smooth sphere. This effect, being a manifestation of designed resonances with higher quality factors, is responsible for elevating the total scattering cross-section. The delay between specular return and creeping wave reflection is 0.36 nsec if peaks of

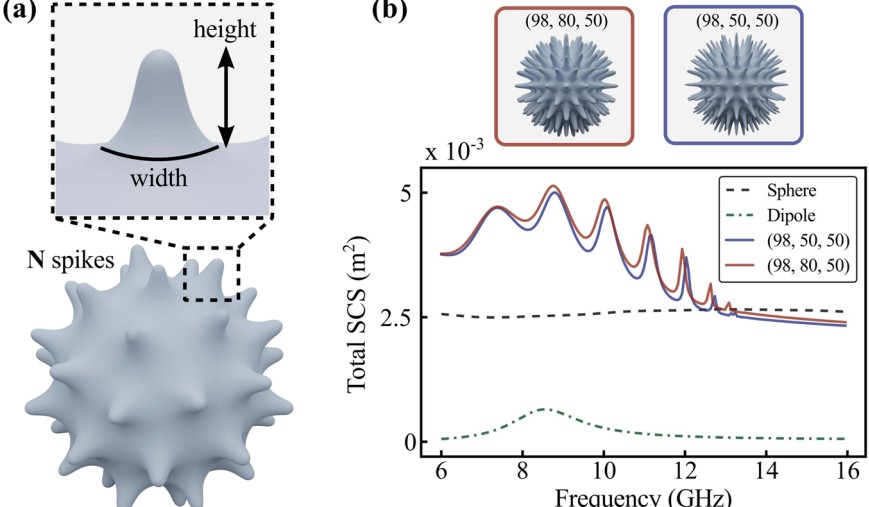

**Fig. 1 Coronavirus-like scatterer. a** Geometry and parameters for optimizing corona scatterers. N-number of spikes. **b** Numerical results: total scattering cross-section (SCS) spectra for a smooth sphere, two optimized corona scatterers, and a half-wavelength dipole. A plane wave (E-electric field, H-magnetic field, k-vector of propagation) is incident on the structure at any angle.

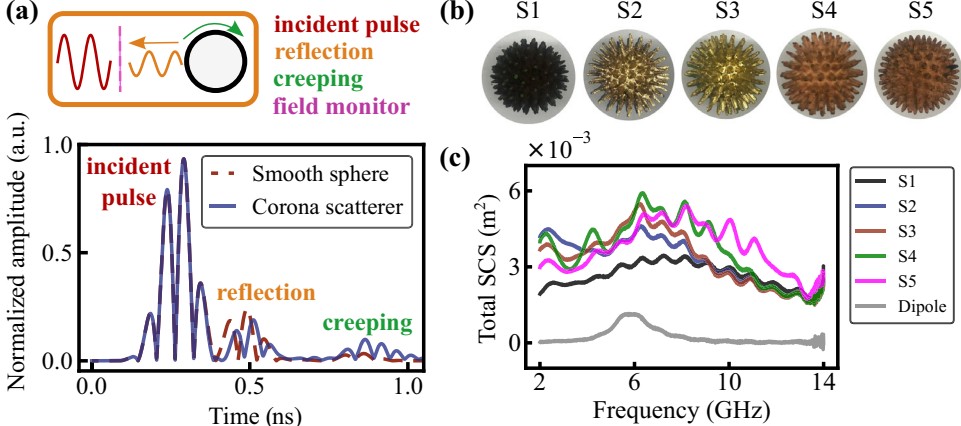

**Fig. 2 Creeping waves analysis and fabrication of samples. a** Creeping waves analysis. Upper insets—schematics of the phenomenon. The main plot—time-dependent back scatting (normalized pulse amplitude, absolute value) for a smooth sphere and the corona scatterer. **b** Photographs of the fabricated samples. Fabrication routines are elaborated in the main text. **c** Total scattering cross-section spectra (SCS) of the samples.

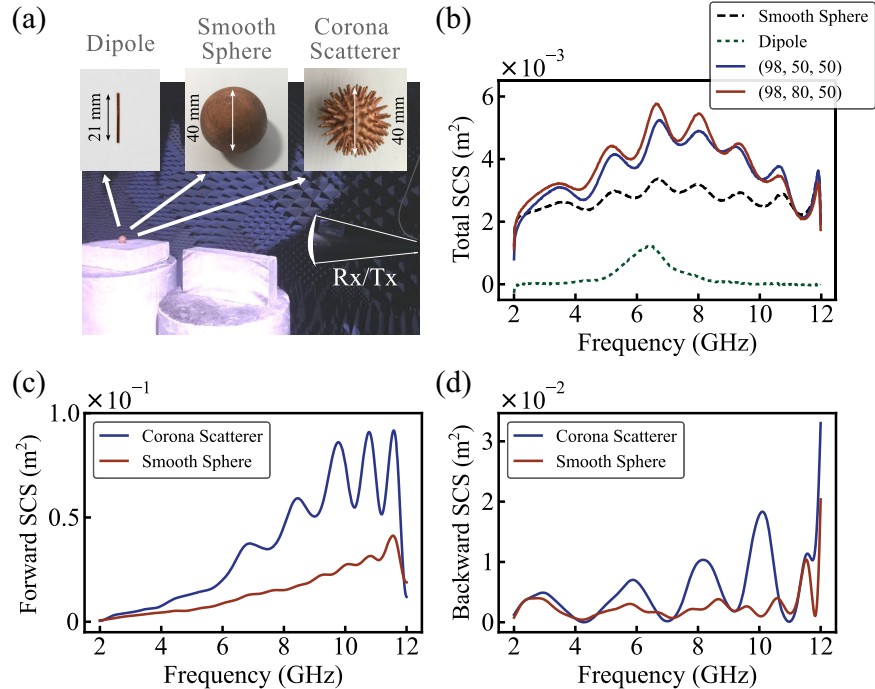

**Fig. 3 Experimental analyze. a** Experimental setup. Anechoic chamber with the receive and transmits antennas (Rx and Tx) and the samples (half-wave dipole, smooth sphere and "corona"-like sphere) with size. **b** Total scattering cross-section (SCS) from all samples. Red and blue curves—"corona"-like spheres, black dashed curve—smooth sphere, green small dashed curve—half-wave dipole. **c** Forward and (**d**) backward scattering cross-sections. Red curves—"corona"-like sphere with highest total scattering (98,80,50), blue curves—smooth sphere.

envelopes are considered. The distance deference is $\Delta d = 105$ mm which corresponds to $\pi r + 2r$, where $r$ is the radius of the sphere. This factor (half of the circle circumference + the diameter) is calculated from the geometrical phase delay between the specular reflection and the creeping wave. Other examples, verifying the contribution of creeping waves experimentally, appear in Supplementary Note S4.

### Experimental measurements

*Total scattering cross-section.* Total scattering cross-section spectra are measured by sweeping the frequency and collecting the forward scattering with the Rx antenna. The optical theorem is applied to obtain the total scattering cross-section via post-processing[26]. The results appear in Fig. 3b and are well-comparable with the theoretical predictions (Fig. 1b). Corona scatterer significantly outperforms the resonant dipole by means of the total scattering cross-section and the bandwidth.

*Backward and forward cross-sections.* In several applications, including interrogation of a scene with a monostatic radar, scattering in a backward direction has value. Figure 3c, d summarizes the results. Furthermore, the backward scattering of the spiky device is significantly higher, compared to the sphere. The wavy behavior of the spectra corresponds to resonant creeping waves.

*Creeping waves.* To observe the creeping waves, the time-domain scenario was reproduced with measurements of complex reflection coefficients ($S_{11}$ parameters). Since the system under study is linear, it can be equally assessed at either frequency or time domains. In this case, the incident pulse undergoes spectral decomposition, each frequency is propagated independently, and the time domain result is reconstructed with an inverse Fourier transformation. For accurate quantitative measurement, a calibrating metal mirror was located at the 1 m

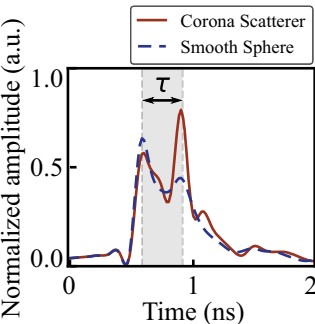

**Fig. 4 Interrogation of scatterers with an ultra-wideband monostatic radar.** Sphere—blue dashed line, corona scatterer—red dash-dotted line.

distance, and the reflected signal was detected. Next, the sphere was placed at the same distance, $S_{11}$ parameters were retrieved, and the time-domain response was reconstructed. One can observe the first specular return and the weak second peak, delayed by 0.33 ns (Fig. 4, blue dashed). The distance between the peak (speed of light over the delay) is $\Delta d = 100$ mm which corresponds to $\pi r + 2r$, where $r$ is the radius of the sphere. Interrogation of the corona scatterer demonstrates a significantly enhanced contribution of the creeping wave (Fig. 4—red solid line). Nearly the same time delay of 0.33 ns was observed, confirming the fair comparison between the structures. The experimental results recover the envelope response and do not resolve the carrier wave oscillations (seen in Fig. 2a, theory). However, the creeping waves amplitudes ratio between the corona scatterer and smooth sphere remains nearly the same both in the experiment and in the theory. The factor here is about 2. Experimental results on other common structures appear in Supplementary Note S4, where the backscattered pulse splitting is revealed.

*Multipole expansion.* To reveal the operation principle of the corona scatterer, the multipole expansion of its scattering cross-section will be performed next. Scattering patterns are well described with a fast-convergence series of multipoles in the case when small structures are considered. Furthermore, in a case of a compact superscatterer, several multipoles are spectrally over-lapping, constructively contributing to the cross-section. Figure 3b demonstrates that the corona scatterer prevails over the resonant dipole in terms of the total scattering cross-section and the scattering bandwidth, which motivates considering the impact of higher-order multipoles.

There are a few possible multipole expansions for analyzing electromagnetic scattering and radiation problems[2,27–29]. They have been applied to various systems and obviously provide identical scattering patterns. The choice of a particular decom-position is mainly related to the results' interpretation and benefits for the analysis of particular effects. Cartesian multipole expansion[29–32] will be used here (technical details appear in Supplementary Note S6). Figure 5 is the summary of the results. Few observations can be done. First, the multipole expansion converges to the total scattering quite well in the frequency band

between 6 and 8 GHz. Several key contributing multipoles resonate at this range, elevating the scattering cross-section quite significantly, compared to a single resonant dipole. Secondly, for higher frequencies, the multipole expansion, encompassing 6 basic multipoles, does not converge to the total scattering cross-section (~x2 mismatch for 12 GHz). Consequently, much more terms in the expansion are needed. This is a typical case, where electrically large scatterers are considered[33]. Those scenarios, however, are better described with the aid of physical optics. Since the scatterer's physical size does not change, it operates as an electrically small object at low frequencies, while becoming wavelength-comparable and large for higher frequencies. Hence, the extended scattering bandwidth of the corona element originates from a combination of a resonant cascading effect (low frequencies) and ray optics (creeping wave management) for high frequencies.

*Near-field distributions.* Finally, accurate near-field scans have been performed to verify a strong accumulation of energy around the corona spikes. The sample was illuminated from the horn antenna and the field is scanned with a near-field probe—a non-resonant dipole, polarized along the incident field. The scheme of the scanning planes is presented in the insert in the upper left corner of Fig. 6. Panels on the figure demonstrate different cuts (a) and (b), which present different maps of scanning planes. Experimental and numerical results are placed side by side. Figure 6a1, b1 shows simulation result and 6(a₂), 6(a₃), 6(a₄) experimental results on (a) cutting panel, 6(b₂) experimental results on (b) cutting panel. Remarkable similarities can be observed. A significant near-field accumulation appears around the spikes and governs the scattering performance. The experiment is performed at 8 GHz, where both resonant multipoles and wave interference phenomena take place.

## Conclusions

Broadband, all-angle all-polarization corona scatterer has been developed and demonstrated. Topology optimization of spikes, equally distributed on a spherical surface, has been performed to maximize both the total scattering cross-section and the band-width. The final design was shown to outperform a resonant dipole by almost an order of magnitude both in the cross-section and the scattering bandwidth. The broadband operation of the small-size structure is attributed to its combined response. While for lower frequencies, its response is governed by cascaded

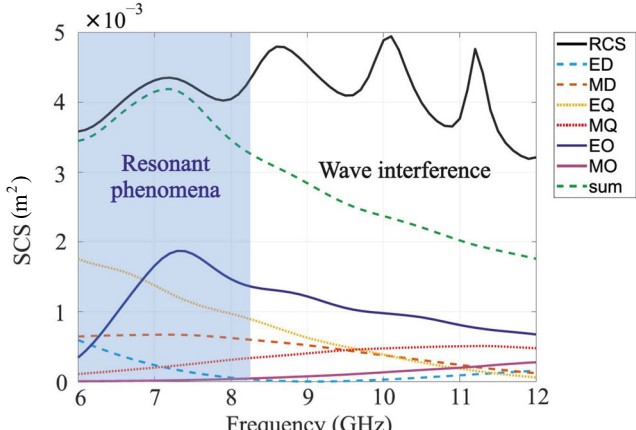

**Fig. 5 Multipole expansion of the scattering cross-section (SCS) of corona scatterer.** The electric (ED) and magnetic dipole moments (MD), EQ, and MQ are the electric and magnetic quadrupoles, EO and MO electric and magnetic octupoles, and the sum of all multipoles is indicated in the legend.

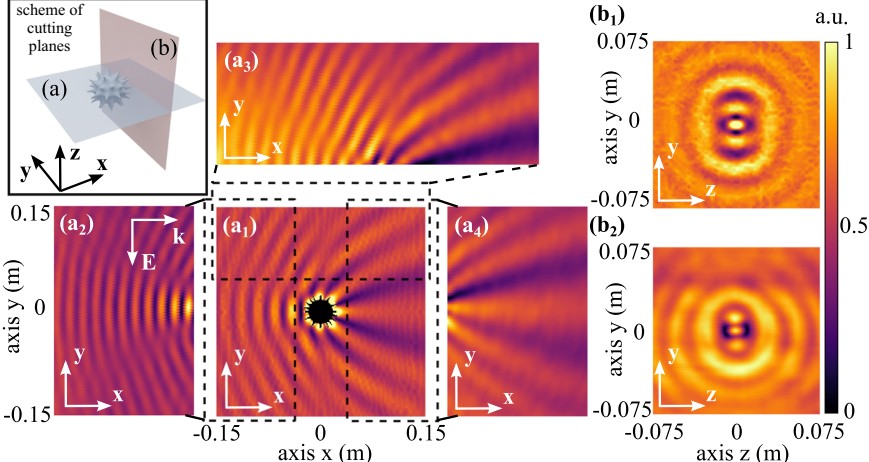

**Fig. 6 Near-field scanning.** The scheme of the cutting planes is presented in the insert in the upper left corner. The field distributions at xy-plane (**a**) and yz-planes (**b**). E-field polarized along y and propagation along x. Numerical results (a₁ and b₁) and measurements (a₂–a₄ and b₂). The frequency of the incident wave is 8 GHz.

constructively interfering resonant multipoles, tunable accumulated phase of creeping waves, traveling along the circumference of the device, builds up the scattering cross-sections for higher frequencies in the spectrum. This combined response allows for covering 6–12 GHz scattering bandwidths.

Additive manufacturing methods, followed by polymer metallization, were developed and used to manufacture the samples. Three different metallization methods, including electroless deposition, magnetron sputtering, and electrodeposition of copper were implemented and all demonstrated reliable electromagnetic performances.

A new generation of efficient scatterers, immune to spatial orientation, could find use in numerous wireless applications, including radar chaff, navigation beacons, alignment marks, and many others.

## Methods

**Simulation and optimization.** Our search space is formulated as follows: the overall number of spikes ($N$, $N < 98$ was considered owing to practical constraints of the subsequent fabrication), the spikes are centered over vertices, distributed on the spherical surface uniformly (the topological definition will be given hereafter). The radius of a virtual sphere, enclosing the entire structure, is 20 mm. A series of spikey spheres (corona scatterers) was created to conduct numerical experiments. The main parameters of the structure are the height of the spikes, their width, and the overall number of spikes ($N$). We used Goldberg polyhedron[34]$\{3, 5 + \}_{N,0}$ for $N = 1,2,3$ was 12, 42, and 98. This construction approximates a spherical surface with a convex polyhedron with three properties: each face is either a pentagon or hexagon, exactly three faces meet at each vertex, and they have rotational icosahedral symmetry. A detailed description of the construction, including relevant codes, appears in Supplementary Note S2.

In our case, each optimization step requires performing a full-wave analysis of a wavelength-comparable resonant within a broad range of frequencies. CST microwave studio with frequency solver was used for full-wave electromagnetic simulation. Since this computation is rather time-consuming (each model has $9 \times 10^5$ mesh cells and runs ~21 h on 512 GB RAM 3.3 GHz) several restrictions on the topology are applied. Blender 3D modeling software package has been used to realize the construction[35]. The centers of mass of polygons (pentagon or hexagon in Goldberg construction) are the spike locations. For the summary, the construction algorithm is: (1) Goldberg polyhedron is created, $N$ defines the number of spikes, (2) a center of mass of each polygon is lifted up perpendicularly to the surface and then connected to each vertex, creating a pyramid, (3) pyramids are smoothed with SubSurf modifier (Blender software), which creates a 2D spline surface. The width of a spike varied from 0.02 to 0.08 with a 0.03 step and the height changed from 0.1 to 0.5 with a 0.2 step. Those ranges were selected manually to comply with fabrication restrictions. Figure 1a demonstrates a typical example. Finally, 27 models with different parameters were selected for a numerical assessment. As the convex enclosure of all the structures is the nearly same, a fair comparison of scattering cross-sections can be performed. It is worth noting that an Integral Solver, considering surfaces made of perfect electric conductors, might solve the problem more efficiently at an expense of neglecting material losses, which are significant in case of near-field accumulation in the vicinity of the structure.

**Metallization.** Since the resulting geometry of the corona scatterer is rather complex, standard fabrication techniques, e.g., milling or casting, are rather involved both in terms of effort and costs. Furthermore, the overall weight of the device is quite high

(we made a test tin sample by casting, which resulted in an overall object's weight of 235 g). Considering those engineering constraints, we performed the deposition of metals on a 3D-printed plastic skeleton. Three different techniques have been applied and the electromagnetic properties of the samples were assessed. Photographs of the samples appear in Fig. 2b and correspond to the following fabrication technique—S1—Methodology 1: electroless deposition, S2, S3—Methodology 2: magnetron sputtering, S4, S5—Methodology 3: electrodeposition of copper.

*Methodology 1.* Electroless deposition of copper on Polylactic Acid (PLA) plastic skeleton was performed by the method described in ref. [36]. Prior to copper deposition, the samples were sequentially washed with deionized water for 5 min, with methanol at a temperature of 50 °C for 5 min, N, N-Dimethylformamide at a temperature 50 °C for 5 min, ethyl alcohol (96%) for 5 min, and then deionized water for 5 min. The next step is a sensibilization in a solution, containing $SnCl_2$—70 g/l and HCl—40 ml/l for 30 min at room temperature, followed by deionized water rinsing. The next step, activation of the surface with Pd-citrate solution (0.1 g/l of $PdCl_2$, 7.35 g/l of citric acid, 2.8 g/l of NaOH, and 2.4 g/l of HCl) at room temperature for 1 min, followed by deionized water rinsing. The last stage is the deposition of copper on the surface of PLA plastic from a solution consisting of $CuSO_4$ 15 g/l, K-Na-tartrate 30 g/l, $Na_2CO_3$ 10 g/l, NaOH 40 g/l, by adding formaldehyde to it to a concentration of 35 vol%.

*Methodology 2.* The gold coating was applied by magnetron sputtering. Prior to an Au deposition, a Cr coating with a thickness of 30 nm was deposited on the PLA sample to improve the adhesion of gold, after which a gold layer with a thickness of 100 nm was deposited on the prepared sample.

*Methodology 3.* Electrodeposition of copper[37,38] was performed in a two-electrode cell consisting of a corona scatterer (PLA plastic skeleton was 3D-printed on a standard FLASHFORGE ADVENTURER 3 device) as a working electrode and a copper plate as a source of $Cu^{2+}$ ions. The sample was pre-washed in EtOH to improve wettability and remove impurities. To create a conductive coating, the surface of the sample was covered with graphite glue (GRAPHIT 33 (KONTAKT CHEMIE, Switzerland)) in three layers. After applying each layer, the samples were placed in a drying oven at a temperature of 60 °C for 30 min. An aqueous electrolyte for copper electrodeposition was composed of 100 g of $CuSO_4*5H_2O$ (Sigma-Aldrich) 35 ml of concentrated $H_2SO_4$ (Sigma-Aldrich), 10 ml of ethanol for reducing surface tension, and 0.05 mg of gelatin as an inhibitor. Electrodeposition of copper was carried out by applying a direct current with a DC power supply SPS-606 (GW Instek, Taiwan). The current strength for copper deposition was set at 1 A and the electrolyte was stirred using a magnetic stirrer at 300 rpm over 4 h.

*Samples comparison.* Total scattering cross-sections of the samples were measured in an anechoic chamber (details will come next) and appear in Fig. 2c. While most of the samples demonstrated similar results (~20% differences), the electrodeposition of copper was found to provide the best performance. Furthermore, from a technological standpoint, this method has numerous advantages, including fabrication cost, reliability, and robustness. Those samples will be used for further experimental analysis. Detailed analysis of surfaces and further discussion of the methodologies appear in Supplementary Note S3.

**Measurement.** The photograph of the experimental setup for scattering tests appears in Fig. 3a. It consists of the receive and

transmits antennas (Rx and Tx), and the sample under test (SUT) located at the antenna far-field. The antennas are broadband (2–20 GHz) NATO IDPH-2018 horns. Horns are connected to N5232B PNA-L Microwave Network Analyzer 300 kHz–20 GHz (PNA—Performance Network Analyzer) ports and calibrated. The half-wave dipole (21 mm), smooth sphere (radius 20 mm), and two corona scatterers were used for the measurements as SUT. Four different measurements have been performed.

## Data availability

All data generated or analyzed during this study are included in this published article. The modeling script is available online: https://zenodo.org/record/6962448#.ZA8jvS8w2Lc.

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

## Acknowledgements

The research was supported by the Department of the Navy, Office of Naval Research Global under ONRG award number N62909-21-1-2038. V.B. acknowledges RRF project Latvian Quantum Technologies Initiative Nr. 2.3.1.1.i.0/1/22/I/CFLA/001 and A.P. acknowledges the Royal Academy of Engineering under the Research Fellowship scheme.

## Author contributions

P.G. supervised the project and revised the paper. A.M., D.V., D.D and K.G. were responsible for the design, optimization, characterization and data analysis. K.L., V.B., A.P. and P.B. participated in the research. D.S.K. fabricated samples. All the authors contributed to discussions and writing of the manuscript.

## Competing interests

The authors declare no competing interests.
