## [Peer Review File · Communications Engineering]

Coronavirus-like All-angle All-Polarization Broadband ScattererReviewers' comments:

Reviewer #1 (Remarks to the Author):

This is an interesting paper on the significant enhancement of the total scattering cross section (scs) generated by a so-called "corona-type" scatterer. The effect of creeping waves traveling around the scatterer is thoroughly investigated. Then, these waves are utilized to the benefit of enhancing the total scs for a substantial interval of frequencies as well as for all angles and all types of polarizations. Comparisons are performed versus a resonant dipole and a regular (smooth) sphere. Experimental structures have also been realized and are presented in the paper demonstrating the anticipated performance.

Publication of a revised version of the paper can be recommended in which the authors address the following points.

1. How is the "all-polarization" argument justified? There is no systematic analysis of the polarization type of the incident waves. Is it just the expectation that the discussed structures outperform the regular spheres which show "all-polarization" behavior?
2. Lines 113-114: please explain more the pattern by means of which you obtain the geometries of the structures. You provide a reference but it is worth to be analyzed in more details in the paper.
3. Figure 1: explain for which polarizations of the incident waves and for which angles you obtain these results. What will happen if you change the polarization (cf. point 1 above)?
4. Lines 290-291: which is the "Cartesian multipole expansion"? Some details would be useful.
5. Line 295: the term "weak convergence" is misleading here. It has to do with function spaces to which the examined expansions are considered. It would be clearer to state just that more terms are required in the expansions in these frequency regimes.
6. A discussion on sensitivities is needed. To which extent are the manufactured scatterers prone to geometrical imperfections? What happens then? How is the total scs affected for all frequencies, angles, polarizations? At least a couple of the discussed structures need to be analyzed to this direction.

Finally, please correct the following typos and phrases:

- a. Line 38: "geometries"
- b. Line 74: "scatterers"
- c. Lines 248-249: this sentence is not so clear. Which remains quite significant?

Reviewer #2 (Remarks to the Author):

Following are my comments on the manuscript :

The manuscript looks interesting and the results are well explained. However, the authors should address the following points:

1. The motivation towards investigating this particular configuration, viz. a spherical structure with "spikes" should be clearly explained.

2. Computation of creeping wave propagation constants around such a structure must be explained.

3. The sphere may probably be represented as one with an impedance boundary condition if the heights of the spikes are small compared with the sphere radius. Can the authors comment on the scattering characteristics from the configuration under such conditions? A parametric variation of the scattering with the height of the spikes may be provided.

Reviewer #3 (Remarks to the Author):

This review regards the article "All-angle All Polarization Broadband 'Corona' Scatterer" (COMMS-22-0317). The submitted work proposes a novel scatterer which takes a "functionalized" sphere shape. I especially appreciate this work, since it is a combination of engineering (i.e., numerical simulations and topology optimization), physical understanding (i.e., specular reflection and creeping waves), and rigorous experimental measurements (i.e., far field and near field characterization). The manuscript is well-written and well-organized. I fully support the publication of this work in "Communications Engineering". Please see the minor comments listed below.

On Page 3, line 93, I would suggest the use of "topology optimization" instead of "topological optimization".

On Page 3, line 95, "converging" to "convergence".

On Page 3, line 97, "as an initial point to start the optimization" to "as an initial point of the optimization".

On Page 4, line 113 – line 114, please double check the sentence "To distribute N spikes, ...".

On Page 4, the paragraph from line 117 to line 135 discusses a construction algorithm for the intended optimization. For the time being, the presented construction algorithm is more like a scan, as the authors mention on line 130 – line 132 that "the width of a spike varied from 0.02 to 0.08 with a 0.03 step and the height changed from 0.1 to 0.5 with a 0.2 step". That is, please be clearer about which parameter (maybe, the number of spikes?) is being optimized.

Also, in the same paragraph, the authors mention the use of CST – frequency domain solver. As far as I understand, the frequency domain solver is based on a Differential Equation (DE-) based method. The method discretizes the entire simulation domain (which must be truncated by some artificial boundaries, e.g., the perfect matching layers). Thus, it results in a very high amount of cells and thus an exceedingly long computational time. I would suggest the authors to consider Integral Equation (IE-) based methods, e.g., the boundary element method (BEM) or the volumetric method of moments (V-MoM) in their future research. These methods only discretize the boundary or the body of the scatterer and thus can save a significant amount of computational time for each iteration.

On Page 6, line 156, "time a domain" to "a time domain".

On Page 6, line 156 – line 157, I would suggest to use "launching" and "observing" both in the past tense to be consistent with "performed".

On Page 6, in the paragraph from line 156 to line 174, please put a space between a number and a unit, e.g., "20nsec" to "20 nsec".

On Page 8, line 194, what is PLA? Is it Polylactic Acid? Please define the acronym before use.

On Page 9, line 249, "worth noting" to "it is worth noting".

On Page 10, for Fig. 3 (line 254 – line 256), please increase the contrast for Fig. 3(a). At this moment, Tx and Rx antennas are almost not visible.

We appreciate the comments of the Referees, which allowed us to improve the manuscript.

Reviewer #1:

Dear Referee,

Following Your recommendations, we included quite a few additional pieces of information in the new Supplementary. We started to introduce the changes into the main text and it grew significantly above 5k words. We absolutely agree that the results should be made reproducible and, hence, all technical details must be included. We did it in the supplementary, which encompasses 7 pages (a small additional manuscript itself), which cannot be accommodated in the main text. The supplementary is attached to the main submission now.

1. How is the "all-polarization" argument justified? There is no systematic analysis of the polarization type of the incident waves. Is it just the expectation that the discussed structures outperform the regular spheres which show "all-polarization" behavior?

Our reply: By design, the structure has very high rotational and reflection symmetry, suggesting its all-angle-all polarization response. Initially, we made several experiments and did not notice any significant dependence. Following the Referee's suggestion, we made an accurate analysis and introduced this graph in the new Supplementary Information file. We also added a quantitative assessment for the statement in the revised version. (The electromagnetic design and structure's optimization page 5).

2. Lines 113-114: please explain more the pattern by means of which you obtain the geometries of the structures. You provide a reference but it is worth to be analyzed in more details in the paper.

Our reply: Following the recommendation, we introduced a detailed description of the process in the new Supplementary Information. In order to keep the logical details clear, we believe taking the figures away from the main text will be more convenient for a reader (The electromagnetic design and structure's optimization page 4 and Supplementary).

3. Figure 1: explain for which polarizations of the incident waves and for which angles you obtain these results. What will happen if you change the polarization (cf. point 1 above)?

Our reply: We updated Figure 1. (page 6)

4. Lines 290-291: which is the "Cartesian multipole expansion"? Some details would be useful.

Our reply: we included the following in the methodology part in the new Supplementary Information and added references (page 12).

5. Line 295: the term "weak convergence" is misleading here. It has to do with function spaces to which the examined expansions are considered. It would be clearer to state just that more terms are required in the expansions in these frequency regimes.

Our reply: We absolutely agree and introduced the clarification on the convergence, introducing estimates (page 12). <https://www.sciencedirect.com/science/article/pii/S0022407320300674#fig0005>

6. A discussion on sensitivities is needed. To which extent are the manufactured scatterers prone to geometrical imperfections? What happens then? How is the total scs affected for all frequencies, angles, polarizations? At least a couple of the discussed structures need to be analyzed to this direction.

Our answer: Following the recommendation, we introduced several changes in the main text and added 2 sections to the Supplementary Material (page 9).

- Finally, please correct the following typos and phrases:
 - a. Line 38: "geometries"
 - b. Line 74: "scatterers"
 - c. Lines 248-249: this sentence is not so clear. Which remains quite significant?

Our reply: Thank you – we corrected the text.

Reviewer #2:

1. The motivation towards investigating this particular configuration, viz. a spherical structure with "spikes" should be clearly explained.

Our Reply: The main motivation is to develop all-angle all-polarization device with a high scattering cross-section (better than a sphere). In this endeavor, exploring 'a modified sphere' is quite appealing. While there are many possible configurations to explore, a sphere with spikes, which underwent a topology optimization is a promising one. On the practical side, it complies with experimental capabilities of 3D printing. Hence, we have chosen this layout. Positioning spikes is not a trivial problem, considering the curved geometry of the sphere. Quite a few efforts were made in the field of differential geometry on distributing N points on a sphere with the maximal symmetry. There are several possible solutions, considering different metrics. We choose one of them and upon this arrangement we have implemented the topology optimization of the spikes. Following the recommendation, we have introduced the discussion in the text (introduction page 3).

2. Computation of creeping wave propagation constants around such a structure must be explained.

Our Reply:

Creeping waves are not eigen modes of the structure, nevertheless, they are used as a descriptive tool to explain returns from radar targets. In smooth geometries, the resonant reflection is typically related to the circumference of the scatterer. In the case of complex topology, the group delay can appear owing to structuring. Our geometry is relatively small, hence the direct link between surface wave (on an unwrapped planar structure) dispersion and actual delay (pulse splitting in the reflection – direct reflection vs a creeping wave) is approximate. Following the recommendation, we introduced the formula for estimation + description in the revised version. We also added results of measurements of creeping waves, related to other geometries – those appear now in Supplementary (page 7).

3. The sphere may probably be represented as one with an impedance boundary condition if the heights of the spikes are small compared with the sphere radius. Can the authors comment on the scattering characteristics from the configuration under such conditions? A parametric variation of the scattering with the height of the spikes may be provided.

Our Reply: We absolutely agree – following the surface equivalence principle, any finite-size scatterer can be replaced by an impenetrable body with a certain surface impedance. The only practical question is whether it is realizable and whether can it be implemented practically with a simpler layout. Following the recommendation, we introduced a discussion on the revised. To underline the impact of spikes, we performed an additional parametric study, which appears in the Supplementary information. (The electromagnetic design and structure's optimization page 5).

Reviewer #3:

On Page 3, line 93, I would suggest the use of “topology optimization” instead of “topological optimization”.

Our Reply: We absolutely agree and changed.

On Page 3, line 95, “converging” to “convergence”.

Our Reply: We absolutely agree and changed.

On Page 3, line 97, “as an initial point to start the optimization” to “as an initial point of the optimization”.

Our Reply: We absolutely agree and changed.

On Page 4, line 113 – line 114, please double check the sentence “To distribute N spikes, ...”.

Our Reply :We absolutely agree and changed.

1. On Page 4, the paragraph from line 117 to line 135 discusses a construction algorithm for the intended optimization. For the time being, the presented construction algorithm is more like a scan, as the authors mention on line 130 – line 132 that “the width of a spike varied from 0.02 to 0.08 with a 0.03 step and the height changed from 0.1 to 0.5 with a 0.2 step”. That is, please be clearer about which parameter (maybe, the number of spikes?) is being optimized.

Our Reply:

Following the Referee's recommendation, we introduced a detailed description of the process in a new supplementary Information. We introduced the link and clarifications in the revised version (The electromagnetic design and structure's optimization page 4 and Supplementary).

2. Also, in the same paragraph, the authors mention the use of CST – frequency domain solver. As far as I understand, the frequency domain solver is based on a Differential Equation (DE-) based method. The method discretizes the entire simulation domain (which must be truncated by some artificial boundaries, e.g., the perfect matching layers). Thus, it results in a very high amount of cells and thus an exceedingly long computational time. I would suggest the authors to consider Integral Equation (IE-) based methods, e.g., the boundary element method (BEM) or the volumetric method of moments (V-MoM) in their future research. These methods only discretize the boundary or the body of the scatterer and thus can save a significant amount of computational time for each iteration.

Our Reply: We absolutely agree and introduced the clarification. The volume mesh was initially used to accurately address the losses. Since our layers have very high conductivity, we were better to use IE. (The electromagnetic design and structure's optimization page 5)

On Page 6, line 156, “time a domain” to “a time domain”.

On Page 6, line 156 – line 157, I would suggest to use “launching” and “observing” both in the past tense to be consistent with “performed”.

On Page 6, in the paragraph from line 156 to line 174, please put a space between a number and a unit, e.g., “20nsec” to “20 nsec”.

On Page 8, line 194, what is PLA? Is it Polylactic Acid? Please define the acronym before use.

On Page 9, line 249, “worth noting” to “it is worth noting”.

On Page 10, for Fig. 3 (line 254 – line 256), please increase the contrast for Fig. 3(a). At this moment, Tx and Rx antennas are almost not visible.

Our answer: Thank you for the comment. We change Figure 3 in the main text.

REVIEWERS' COMMENTS:

Reviewer #1 (Remarks to the Author):

Thank you very much for addressing convincingly all the points raised in the initial report and for preparing a very detailed supplementary file. The paper can now be accepted for publication.

Reviewer #2 (Remarks to the Author):

The authors have satisfactorily addressed my comments.

Reviewer #3 (Remarks to the Author):

The authors have answered all the questions from the first-round review. Thus, I recommend the publication of the paper.

Reviewer #4

The paper entitled “All-angle All Polarization Broadband ‘Corona’ Scatterer,” in its current format considers both the computational modeling and experimental measurement. In this paper, the authors claim to provide new electromagnetic scatterers utilizing the concepts of resonant conditions as well as physical optics. The frequency range of the analysis is 2 GHz – 12 GHz, which corresponds to the free-space wavelength of 15 cm – 2.5 cm. The diameter of the sphere chosen in this study is 4 cm. For the given frequency range and the size of the scatterer, it is not advisable to choose physical optics. If the authors have opted for physical optics, they must compare the results with the full-wave electromagnetics and comment on the accuracy.

By reading the paper, I come to the impression that the paper in its current format qualitatively represents a research report where the authors systematically ignore the important details. For example, the time-domain numerical method has been used for demonstrating the contribution of creeping waves (Section III, page 6, line 156). However, the details of the numerical method and underlying assumptions, for example, time step, and spatial discretization if the numerical method is finite-difference time-domain (FDTD). Moreover, the boundary condition to truncate the computational domain is missing. Furthermore, the authors have used an ‘optical theorem’ to estimate the total scattering cross-section (line # 240) – I do not understand how this theorem works. On page 11, line # 290, the authors state the use of “cartesian multipole expansion” without explaining it. Similarly, the authors have used optimization techniques for generating corona spikes on a smooth sphere, however, I do not see a clear mathematical derivation or numerical scheme.

General comments:

1. In the abstract, the authors use ‘corona virus-like spikes’. Please remove the word ‘virus’ because this does not sound good in this paper.
2. On page 4, lines 130-131, the width of the spikes varied from 0.02 to 0.08. May I know the unit?
3. On page 4, the authors used ‘Blender software’, however, the objective is not clear to me. Please explain the use of the software clearly.
4. On page 5, line 136, the authors claim the best use of two results – please explain how they are best.
5. On page 5, line 138, the authors used a half-wave dipole antenna for comparing the scattering spectra. Is this a single dipole or an array of dipoles? Why do you use dipole in this study?
6. In section V, line 241, the authors claim that the experimental results are comparable to the theoretical predictions. To the best of my knowledge, no theoretical results are presented in this paper. Please superimpose the results in the same plot and then compare it.
7. On page 12, line 311, there is a typo in the sentence ‘... a non-resonant dipole, polarized along the incident’. Please replace ‘along’ with along.
8. On page 12, line 297, there is a typo in the sentence “ .. Those scenarios, however, are better described with the air of physical optics”. Please replace ‘air’ with area.

Dear Editor,

We appreciate sending us the Referee#4 follow-up report. We addressed all the points and comments and further improved the manuscript. We will kindly ask you to send the Referee all the replies from the previous round, including the very detailed supplementary, which also addresses the main points in this review. Changes are inserted in the revised manuscript and are indicated by red color.

A. Mikhailovskaya

Dear Referee,

We didn't get your report together with the other 3 inputs and, hence, we had no chance to reply to them all at the same time. We added quite a few new data during the review process (addressing points of Referees 1, 2, and 3) and asked the Editor to forward it. We will be happy to further improve our work in case you find it relevant. We especially appreciate the notes on terminology and analysis details, which we overlooked.

Reviewer #4:

1. The paper entitled "All-angle All Polarization Broadband 'Corona' Scatterer," in its current format considers both the computational modeling and experimental measurement. In this paper, the authors claim to provide new electromagnetic scatterers utilizing the concepts of resonant conditions as well as physical optics. The frequency range of the analysis is 2 GHz – 12 GHz, which corresponds to the free space wavelength of 15 cm – 2.5 cm. The diameter of the sphere chosen in this study is 4 cm. For the given frequency range and the size of the scatterer, it is not advisable to choose physical optics. If the authors have opted for physical optics, they must compare the results with the full-wave electromagnetics and comment on the accuracy.

Our response:

We used full-wave simulation for the entire frequency range to model the scattering scenario. We separated the spectrum into 2 areas – resonant regime and interference regime solely for the intuitive expansionary reason. We absolutely agree with the Referee – physical optics is not the correct terminology to use. The better one may be the interference region. Following the recommendation, we change the terminology in figures and in the text – thanks a lot, it is a very important point, we admit it.

2. By reading the paper, I come to the impression that the paper in its current format qualitatively represents a research report where the authors systematically ignore the important details. For example, the time domain numerical method has been used for demonstrating the contribution of creeping waves (Section III, page 6, line 156). However, the details of the numerical method and underlying assumptions, for example, time step, and spatial discretization if the numerical method is finite-difference time-domain (FDTD). Moreover, the boundary condition to truncate the computational domain is missing.

Our response:

It was also noticed in the previous review round and motivated us to make a supplementary with all the details. We absolutely agree each research step has to be reproducible. Sometimes, the details interfere with the text (it makes it heavy to read), hence all the data is made available in the supplementary. Thanks again, we think we will adopt this approach (making tables as supplementary) for our future work.

3. Furthermore, the authors have used an 'optical theorem' to estimate the total scattering cross-section (line # 240) – I do not understand how this theorem works.

Our response:

This is the rather standard method to deduce the total scattering cross-section from the forward scattering amplitude. Somehow, RF people use the terminology “optical” which sounds misleading. However, it is the most commonly used one, and thus everyone keeps using it. We added a reference and a clarification to the revised version.

In terms of σ_t and \mathbf{f} the optical theorem reads

$$\sigma_t = \frac{4\pi}{k} \text{Im}[\boldsymbol{\epsilon}_0^* \cdot \mathbf{f}(\mathbf{k} = \mathbf{k}_0)] \quad (10.139)$$

Jackson, J. D. Classical Electrodynamics 3rd edn. (Wiley, New York, 1999).

In RF, we can measure the phase and hence, this method is of use, rather than scanning the field around the object (or with an integration sphere in optics, for example).

4. On page 11, line # 290, the authors state the use of “cartesian multipole expansion” without explaining it. Similarly, the authors have used optimization techniques for generating corona spikes on a smooth sphere, however, I do not see a clear mathematical derivation or numerical scheme.

Our response:

We included the following in the methodology part in the new Supplementary Information and added references. We also added clarifications in the text. Please, see the places which we indicated with notes.

General comments:

G1. In the abstract, the authors use ‘corona virus-like spikes’. Please remove the word ‘virus’ because this does not sound good in this paper.

Our response: This title was suggested by the editorial office (Chief Editor of Communications Engineering). We believe it aims on attracting readership. We added “-like” at all the places to prevent confusion. If you consider this point critical, we are not insisting on it, but need to get approval from the Editor.

G2. On page 4, lines 130-131, the width of the spikes varied from 0.02 to 0.08. May I know the unit?

Our response: Following the Referee’s recommendation, we introduced a detailed description of the process in new supplementary Information + clarified in the text.

G3. On page 4, the authors used ‘Blender software’, however, the objective is not clear to me. Please explain the use of the software clearly.

Our response: We introduced the link and clarifications in the revised version. The main steps of the scatter’s construction are summarized in Supplementary Table 1.

G4. On page 5, line 136, the authors claim the best use of two results – please explain how they are best.

Our response: We had 27 models with different parameters. All models were numerically studied, and the selection parameter was chosen - the maximum of total scattering – we introduced this clarification in the text.

G5. On page 5, line 138, the authors used a half-wave dipole antenna for comparing the scattering spectra. Is this a single dipole or an array of dipoles? Why do you use dipole in this study?

Our response: We use a single dipole and now highlighted it in the revised. Scattering limits are typically assessed versus a single small resonant dipole – single channel dipolar limit. In this case, we use both the dipole and a sphere for a more comprehensive comparison. We indicated this point in the text.

G6. In section V, line 241, the authors claim that the experimental results are comparable to the theoretical predictions. To the best of my knowledge, no theoretical results are presented in this paper. Please superimpose the results in the same plot and then compare it.

Our response: We used numerical prediction (we changed theoretical to numerical now). We tried to combine all 8 curves together and are not very happy with the visual result – Fig. 1(b) and Fig. 3(b). It also contradicts our logical structure – we will need to jump into the experiment before we analyze the performance.

G7. On page 12, line 311, there is a typo in the sentence ‘... a non-resonant dipole, polarized alon the incident’. Please replace ‘alon’ with along.

Our response: Thanks, changed!

G8. On page 12, line 297, there is a typo in the sentence “.. Those scenarios, however, are better described with the air of physical optics”. Please replace ‘air’ with area.

Our response: We absolutely agree and changed “air” on “aid”.